# Power Grids and Instrument Transformers up to 150 kHz: A Review of Literature and Standards

**DOI:** 10.3390/s24134148

**Published:** 2024-06-26

**Authors:** Mohamed Agazar, Giovanni D’Avanzo, Guglielmo Frigo, Domenico Giordano, Claudio Iodice, Palma Sara Letizia, Mario Luiso, Andrea Mariscotti, Alessandro Mingotti, Fabio Munoz, Daniele Palladini, Gert Rietveld, Helko van den Brom

**Affiliations:** 1Laboratoire National de Métrologie et d’Essais (LNE), 1 Rue Gaston Boissier, 75015 Paris, France; mohamed.agazar@lne.fr; 2Ricerca sul Sistema Energetico-RSE S.p.A., Via Rubattino, 54, 20134 Milan, Italy; giovanni.davanzo@rse-web.it (G.D.); daniele.palladini@rse-web.it (D.P.); 3Swiss Federal Institute of Metrology METAS, 3003 Bern, Switzerland; guglielmo.frigo@metas.ch; 4Istituto Nazionale di Ricerca Metrologica (INRIM), Str. Delle Cacce, 91, 10135 Turin, Italy; d.giordano@inrim.it (D.G.); p.letizia@inrim.it (P.S.L.); 5Department of Engineering, Università degli Studi della Campania “Luigi Vanvitelli”, Via Roma 29, 81031 Aversa, Italy; claudio.iodice@unicampania.it; 6Dipartimento di Ingegneria Navale, Elettrica, Elettronica e delle Telecomunicazioni, Università degli Studi di Genova, Via Opera Pia 11a, 16145 Genova, Italy; andrea.mariscotti@unige.it; 7Department of Electrical, Electronic and Information Engineering “Gugliemo Marconi” Alma Mater Studiorum—University of Bologna, viale del Risorgimento, 2, 40136 Bologna, Italy; alessandro.mingotti2@unibo.it; 8VSL B.V. (VSL), Thijsseweg 11, 2629 JA Delft, The Netherlands; fmunoz@vsl.nl (F.M.); grietveld@vsl.nl (G.R.); hvdbrom@vsl.nl (H.v.d.B.); 9Department of Electrical Engineering, Mathematics and Computer Science (EEMCS), University of Twente, Drienerlolaan 5, 7522 NB Enschede, The Netherlands

**Keywords:** power system, measurements, instrument transformers, high-frequency distortion, supra-harmonics, power quality, grid impedance, medium voltage

## Abstract

The phenomenon of high-frequency distortion (HFD) in the electric grids, at both low-voltage (LV) and medium-voltage (MV) levels, is gaining increasing interest within the scientific and technical community due to its growing occurrence and the associated impact. These disturbances are mainly injected into the grid by new installed devices, essential for achieving decentralized generation based on renewable sources. In fact, these generation systems are connected to the grid through power converters, whose switching frequencies are significantly increasing, leading to a corresponding rise in the frequency of the injected disturbances. HFD represents a quite recent issue, but numerous scientific papers have been published in recent years on this topic. Furthermore, various international standards have also covered it, to provide guidance on instrumentation and related algorithms and indices for the measurement of these phenomena. When measuring HFD in MV grids, it is necessary to use instrument transformers (ITs) to scale voltages and currents to levels fitting with the input stages of power quality (PQ) instruments. In this respect, the recently released Edition 2 of the IEC 61869-1 standard extends the concept of the IT accuracy class up to 500 kHz; however, the IEC 61869 standard family provides guidelines on how to test ITs only at power frequency. This paper provides an extensive review of literature, standards, and the main outputs of European research projects focusing on HFD and ITs. This preliminary study of the state-of-the-art represents an essential starting point for defining significant waveforms to test ITs and, more generally, to achieve a comprehensive understanding of HFD. In this framework, this paper provides a summary of the most common ranges of amplitude and frequency variations of actual HFD found in real grids, the currently adopted measurement methods, and the normative open challenges to be addressed.

## 1. Introduction

The phenomenon of high-frequency distortion (HFD, often called supra-harmonics) consists of disturbances having significant frequency contents beyond 9 kHz. HFD gained significant interest in recent decades due to its increasing presence in both low-voltage (LV) and medium-voltage (MV) grids [1,2,3,4]. In fact, many renewable energy sources, as well as new electronic loads, are connected to the main power grid through power converters, whose switching frequencies are significantly increasing, leading to a corresponding rise in the frequency of injected disturbances. In particular, the main sources of HFD include energy-saving equipment, power electronics technologies associated with renewable energy sources, electric vehicle chargers, power line communication associated with smart grids, LED lamps, and photovoltaic inverters.

Due to the negative impact of HFD on power quality (PQ) and on the normal operation of the grid, including power loss, heating of grid elements, aging of dielectric materials, and interference with equipment and power line communication technology [5], several challenges arise across the various fields of electrical/electronic engineering on how to deal with them. These challenges include the accurate identification and measurement of HFD, assessing their impact on equipment, understanding the transfer impedance between LV and MV grids, considering partial discharge effects, modeling grid impedance, implementing new standards or updating existing ones, and developing effective methods to mitigate their impact on power networks [6,7,8,9].

In the literature, several papers are focused on the topic of HFD. These research studies are focused on measuring the level of emissions generated by different sources [9,10] and their effects on LV and MV power system components [11], evaluation methods of HFD levels [12,13], propagation through power network impedance [14,15], and mitigation strategies [16,17,18,19].

As regards the specific area of HFD, accurate measurement methods and indices for their quantification have been included into in-force international standards such as IEC 61000-4-7 [5] and IEC 61000-4-30 [20]. In addition, suitable measurement techniques have been proposed in the scientific literature to measure HFD [21,22]. For instance, Digital CISPR [23], Light-QP [24], and matrix pencil methods [25] are some of the proposed methods that provide different approaches to measure HFD, which differ from each other in computational resources, accuracy, and immunity to noise [2,26].

As regards the measurement chain for the monitoring of HFD in MV grids, it must always include transducers to properly scale down the voltage and current to levels fitting with the input stage of PQ analyzers or, more generally, acquisition systems. For MV grids, the widely used transducers are instrument transformers (ITs), such as voltage and current transformers (VTs and CTs). Consequently, it is evident that the ITs’ metrological performances at frequencies higher than 9 kHz strongly impact the overall accuracy associated with HFD measurements [24]. From the standardization point of view, the ITs are covered by the IEC 61869 standard family. The recently released IEC 61869-1 Edition 2 [27] extends the concept of accuracy classes, previously limited to the power frequency, over five different frequency ranges up to 500 kHz. However, the entire standard family lacks indications on test procedures, methods, and specifications of the reference systems used for characterizing the ITs at frequencies different from the fundamental frequency. In this respect, the European research project EMPIR 19NRM05 IT4PQ [28,29], concluded in June 2023, focused on building up the entire metrological framework for the accuracy verification of ITs from power frequency up to 9 kHz [30].

As regards the frequency range 9–150 kHz, a new project has recently started: the European Partnership in Metrology (EPM) 22NRM06 ADMIT [31,32]. The most significant activities that will be carried out within this project are definition of accuracy parameters and related procedures for the performance evaluation of ITs, definition of realistic waveforms, development of new reference systems for the generation of alternating current (AC, 50/60 Hz) or direct current (DC) voltages (up to 36 kV) and currents (up to 2 kA) with spectral components at reduced amplitudes up to 150 kHz, and development of reference MV voltage and current sensors up to 150 kHz. The first key activity to achieve these objectives is the collection of all available information regarding HFD occurring in electrical grids in the 9–150 kHz range and the behavior of ITs in this range. In this respect, this paper presents the outcome of this extensive review activity, summarizing all data obtained from the analysis of scientific literature, international standards, and previous research projects focusing on similar topics. This preliminary study of the state-of-the-art represents an essential starting point for defining significant waveforms to test ITs and, more generally, to achieve a comprehensive understanding of HFDs and how to measure them.

The paper is organized as follows: Section 2 provides an overview of data from the scientific literature, focusing on amplitude and frequency characteristics, measurement techniques, propagation, and mitigation methods of HFDs. Section 3 offers a review of related standards, and Section 4 presents a review of previous research projects. Section 5 summarizes the main findings, while Section 6 draws the conclusions.

## 2. Scientific Literature Review

This section presents a comprehensive analysis on HFD drawn from a deep review of scientific literature. As already anticipated in the Introduction, from the scientific literature it is evident that the HFDs are mainly injected in power grids due to the integration of generation systems based on renewable energy sources into the electric grid using static power converters [1,2]. Considering the different nature of HFDs, their amplitudes and frequencies can span wide ranges. Therefore, Section 2.1 aims to identify the typical occurrence of amplitudes and frequencies within the 9–150 kHz range commonly observed in electric grids. The sections from 2.2 to 2.4 summarize information from the scientific literature dealing with HFD measurement techniques, the influence of grid impedance on HFD propagation, and mitigation strategies, respectively.

### 2.1. Typical Amplitude and Frequency of HFDs

The analysis presented in this section involved an exhaustive review of several papers to ensure a comprehensive understanding of the typical amplitude and frequency components of HFD.

The most significant HFD sources were revealed to be photovoltaic (PV) power parks and electric vehicles (EVs) [17] and their charging stations (EVCSs), which are more intimately interconnected in the MV and LV grids, rather than larger wind parks and other large power installations, usually connected farther away in the grid an closer to the high-voltage level, with some exceptions [18,33,34].

In [35], a voltage HFD at 16.1 kHz was observed, in addition to the main HFD peak in the spectrum at the frequency of 2.5 kHz. The occurrences of this HFD indicate that they were caused by PV power installations in the considered grid. In [36], a dominant emission at 10 kHz was found due to the switching frequency of the power converter. The paper underlined the time-varying emissions due to the variation in switching frequency of wind turbine converters at different operating modes during low power extraction, with switching frequencies ranging from 7 to 10 kHz. Therefore, 10 kHz can be considered an HFD frequency component due to wind turbine converters, which can impact the network and connected equipment. In [37], HFD components were observed in the range 2–50 kHz for both voltage and current. Here, the predominant frequencies were 12.5 kHz, 16.1 kHz, and 34.1 kHz, with amplitudes below 0.1% and 0.25% for voltage and current, respectively. In [38], a measurement campaign in a distribution network was performed. The paper showed that the components of HFDs were primarily concentrated below 50 kHz, with amplitudes generally below 0.3 V, particularly around 10 kHz and 12 kHz. Moreover, in the presence of high PV penetration, 10 kHz, 26 kHz, and 36 kHz frequency components were found predominant, compared with tones at other frequencies. Finally, a component with 0.15 V at 126 kHz was found. Probably, this is due to the presence of resonances in the system, as caused by grid-connected inverters and their filters. In [39], a long-term measurement was performed at several locations with small-scale PV installations. The results of the measurements showed that there were two dominating HFD frequencies at 16 kHz and 16.7 kHz. However, no additional information was provided regarding the amplitude of these HFD components.

The study in [40] examined the intermodulation resulting from the interaction between PV systems and electric vehicle (EV) charger stations, demonstrating that it produced HFD at several frequencies with significant amplitudes. In particular, a dominant HFD current frequency was identified at 16 kHz, probably due to the switching frequency of the PV inverter circuit. Additionally, lower-amplitude emissions were observed at integer multiples of the switching frequency, especially at 32 kHz and 48 kHz. Furthermore, the primary emission of the EV charger station was detected below 25 kHz, but during the charging process, a component around 100 kHz emerged, likely attributed to the switching frequency of the EV. The intermodulation between the main emissions of PV and EV generated HFDs at various frequencies, including 16 kHz, 32 kHz, 52 kHz, 48 kHz, 64 kHz, 68 kHz, 80 kHz, 84 kHz, and 96 kHz.

In [41], HFDs generated by Battery Electric Vehicles (BEVs) in a LV network were examined. The paper highlighted that BEVs can produce a high level of HFDs, which can significantly impact PQ. In particular, the findings revealed that the HFD components with higher amplitudes for BEVs charging at 16 A were concentrated around 10 kHz, reaching magnitudes of up to 1080 mA. Harmonics of this component were observed at approximately 20 kHz and 40 kHz. Furthermore, for BEVs charging at 23 A, the dominant HFD components with higher amplitudes were found around 45 kHz, with magnitudes up to 200 mA, and their harmonics were noted at around 90 kHz.

In [42], a measurement campaign conducted in LV grids detected the presence of HFDs within the context of residential PV inverter installations. Specifically, peaks were observed at approximately 16 kHz and 19.5 kHz.

In [43], the transmission of HFD distortions through a transformer from the LV grid to the MV grid was studied. HFD currents were injected using an HFD-emitting device, which can create components in the frequency ranges 3.3–3.9 kHz, 6.8–7.8 kHz, and 10–11 kHz. The paper showed that HFD can propagate with various transfer ratios (i.e., a transfer ratio of 0.5 V/V means that 1 V becomes 0.5 V) for voltage or current and from LV to MV, or vice versa. In particular, from LV to MV, the current ratio was approximately 1 A/A, whereas the voltage ratio was between 0.01 V/V and 1.0 V/V and depended on the transformer ratio. The transfer ratio from MV to LV was, instead, between 0.5 V/V and 3.0 V/V, while no information was reported for the current transfer ratio. In [44], an advanced converter harmonic model was introduced to study harmonic resonance when an ultra-fast charging station was connected to the MV grid. This model was used for the impedance calculation up to 2.5 kHz. In [45], the propagation of HFD in a MV network was studied, focusing on the frequency range of 2–50 kHz. The analysis was based on the on-site measurements carried out in a 20 kV public network, which included a small MV wind plant and solar parks distributed over the whole network and connected to the MV grid via dedicated transformers. The results showed HFDs characterized by amplitudes of tens of millivolt and high variability during the day. In [19,46], the impact of voltage HFDs on MV cables was evaluated. In [46], the accelerated aging of MV cable terminations due to HFDs was addressed in order to provide recommendations to evaluate the risk of cable terminations’ failure under the presence of HFD in MV networks in the frequency range of 2–150 kHz. This paper suggested not to overcome the 24% of the rated voltage under 20 kHz and 9% of the rated voltage above 20 kHz. In [19], the effects of HFD on cable insulation were analyzed. It was shown that cable insulation degradation progresses faster under power frequency voltage with superimposed HFDs. The test conducted in this paper consisted of superposing damped sine wave pulses with a natural frequency of 7.2 kHz and a repetition frequency of 800 Hz to the power frequency voltage. Moreover, the peak-to-peak voltage was set to 15% of the power frequency peak-to-peak voltage. These parameters were chosen according to a numerical simulation, which emulated a 1 MW PV power plant connected to the MV distribution grid.

The HFD distribution in terms of amplitude and frequency was also evaluated, focusing on a wide range of victims in [24], providing a justification for limits in the supra-harmonic range based on documented negative effects. Such negative effects range from the already mentioned impact on MV cables, and cable joints in particular, to aging of capacitors and dielectrics in general, as well as flicker as acoustic noise generated by interference to electrical appliances and EVs themselves, to interference to electrical equipment; in particular, in the presence of electronic circuits and communications, as in the case of power line communication (PLC) technology. Being the base for grid monitoring and automation, protection of PLC devices and communications from interference is a priority [47]. An important part of this wide group of electrical devices is represented by devices for protection (residual current devices) and metering (energy meters). As for protection, the real risk is desensitization of the protection relay, for which in the presence of HFD, the tripping threshold is increased, with relevant implications for electrical safety [48]. Metering is heavily influenced by HFD, as demonstrated in [49], where devices working on different principles but equally available on the market and complying with the same standard reacted quite differently. The problem is not only susceptibility of the electronic devices inside, but the different algorithms used to calculate the active and reactive power, as well as the interpretation of the active and reactive power terms (as demonstrated in [50,51] for railway applications under various amounts and patterns of distortion for rolling stock in 16.7 Hz and 50 Hz systems).

### 2.2. Measurement Methods

Various measurement methods have been proposed for identifying and quantifying the occurrence of HFDs in power grids [2,21].

More specifically, the international standards IEC 61000-4-7 [5], IEC 61000-4-30 [20], and CISPR 16-1 [23] provide methods for measuring HFDs. The proposed method for measuring frequency components above 9 kHz involves aggregating the energy of the signal under analysis into predetermined frequency ranges. Whereas for frequencies up to 9 kHz the commonly employed frequency bandwidth is 200 Hz, a bandwidth of 2 kHz is suggested for higher-frequency components. However, it is important to note that these methods offer reduced computational complexity but may sacrifice accuracy, particularly for high-amplitude components.

However, additional methods based on different techniques have been developed and presented in the scientific literature. These methods provide different accuracy and resolution associated with both amplitude and frequency of HFD [2,21,22,42,52]. The different accuracy and resolution of HFD measurement methods can be attributed to the following causes:The susceptibility to noise: some methods are less immune to noise and may lose accuracy in estimating high-amplitude HFDs.Computational complexity: some methods require more mathematical operations during processing, affecting their efficiency.Measurement equipment: the sampling rate, resolution, and accuracy class of measurement equipment can affect the accuracy of methods.Calibration process and data processing techniques can also contribute to the variation in accuracy and resolution.Transducers and sensors: all of the measurement methods are influenced by the accuracy and precision of the transducers and sensors.

More generally, the developed measurement methods can be categorized into two main groups: time-domain and frequency-domain measurement techniques.

The time-domain methods involve analyzing the waveform using cross-correlation and matrix pencil to process the measured signal and estimate the HFD components. These measurement methods include Digital CISPR [23], Light-QP [24], Numerical-Heterodyne [53,54], the matrix pencil method [22], and the sliding-window TLS-ESPRIT. The advantages of time-domain measurement methods include high time resolution and the ability to effectively reduce the influence of colored noise on the measurement [22,52]. However, time-domain methods have limitations in terms of computational complexity and accuracy in estimating high-amplitude HFD components.

The frequency-domain methods involve analyzing the spectral content of the waveform. These methods use the Fourier transformer (FT) in order to analyze the frequency component up to 150 kHz [2,21,22,52,55]. The advantage of these methods includes the ability to provide wide frequency-domain measurement and higher-frequency resolution. However, these methods are susceptible to noise and require a higher sampling rate to accurately detect the HFD components.

The main issue across all methods concerns the amplitude resolution of measurement devices in the high-frequency range. Typically, the input stage of these devices is designed based on the amplitude of the fundamental component. Consequently, when an HFD with an amplitude below 5% of the fundamental component is present, the resolution of the measured components diminishes compared to the fundamental component.

To address this issue, many measurement approaches include a high-pass filter to strongly attenuate the amplitude of the fundamental component. In this way, the input stage is designed to fit the high-frequency component, optimizing the amplitude resolution for HFDs. However, it is worth noting that scientific literature lacks papers addressing the metrological characterization of sensors, transducers, and high-pass filters specifically developed for MV applications from 20 kHz to 150 kHz.

### 2.3. Propagation of HFDs: Grid Impedance Measurement

The monitoring of HFD propagation in power grids through the grid impedance is a crucial issue since it can cause resonance phenomena. Resonance is common to DC and AC grids, as a consequence of the interaction between the grid impedance itself and the admittance contribution of connected loads, in particular renewables interfaced by means of actively controlled inverters. This is not only a characteristic of AC grids but also quite common in DC grids [56], and can affect not only “grids” intended as distribution networks, but also specialized systems, such as electrified railways [57]. HFD from sources connected to the MV network, such as PV installation, wind farms, EV charger stations, and all renewable sources that use switching power converters, will spread through the MV network and, from there, through the transfer impedance to equipment connected to the LV grid. Furthermore, LV devices, such as LED lamps, can produce HFD and, from there, can be transferred to MV grids [58,59]. Therefore, the equipment connected to both the LV and MV grids can be interfered with or damaged by the propagation of HFDs.

Methods for grid impedance measurement are classified into active, passive, and quasi-passive methods, each with its own advantages and disadvantages [60].

Active methods are the most accurate, as they rely on a test signal injected into the grid. The shape of the test signal may vary, from ideally a pulse to a swept sine, or a combination of sinusoidal components at selected prime frequencies, or a pseudo-random binary sequence. The test signal must be efficiently coupled into the grid with problems of galvanic insulation and bandwidth of the amplifying and coupling device.

Passive methods, instead, are based on passive listening of the electrical quantities and rely on transients and excitation signals that are intrinsic to the grid itself. Such excitation signals may be the spectral components of emissions of distorting loads, as well as transients due to maneuvers (load or compensator switching, breakers tripping, etc.). This approach was successfully used to identify the railway line impedance following electric arcs, exploiting the high-frequency content of the arc transient and compensating the reduced number of samples with advanced spectral estimation techniques [61]. An impact evaluation based on the kind of interference between HFDs and power networks was introduced in [15,16,18]. These papers considered adverse effects, such as audible noise, light flicker, tripping of residual current devices, and cable termination failure.

HFD propagation generally depends on the impedance of the LV and MV grids and the connected devices. The sensitivity of HFD propagation to changes of the grid impedance at the delivery point has been modeled for the case of EV chargers connected to a LV DC distribution grid [14] and assessed in two experimental case studies [43,44]. It has been found that HFDs can propagate through transformers, with different transfer ratios depending on the direction of propagation. Moreover, HFDs can interact with close devices in end-user installations, and the propagation toward the grid can either increase or decrease depending on the ratio between the device and grid impedance [62].

For these reasons, it is evident that a key issue in the HFD framework is the frequency-dependent grid impedance measurement, which provides more accurate information about the connection capacity of points of common coupling, enabling more efficient grid utilization and optimization of large-scale generation plants. To measure grid impedance in the frequency range of 9–150 kHz, several methods have been proposed in the literature:A windowed Fourier transform is a commonly used method to improve the calculation precision and anti-interference ability of grid impedance detection [63].A three-stage interpolation-based measurement method, using cubic spline interpolation and Hanning-window-based interpolated discrete Fourier transform (IpDFT), can remove undesired components and deal with spectral leakage caused by FFT [64].Spectral excitation currents can be used to identify grid impedance, and different measurement systems have been successfully developed for different voltage levels [65].A time–frequency distribution method using a single rectangular pulse injection is employed for grid impedance estimation in the frequency range of HFD [66].Advanced methods for characterizing LV grid access impedance in the frequency range assigned to Narrow-band power-line communications (NB-PLC) have been developed and validated [59].

The main issue of grid impedance measurements is related to the emissions from several sources, such as non-linear loads or power electronics, and varies with frequency. These emissions can corrupt impedance measurements, leading to false resonances, phase distortion, and additional coupling of impedance elements [67]. For these reasons, the identification of accurate techniques for the broad-band measurement of grid impedance, particularly at MV, is still an open issue.

### 2.4. HFD Mitigation Strategies

Mitigation of HFD is a crucial task [5] to avoid equipment malfunctions and decrease of the equipment lifespans and to maintain power grid stability. It is worth pointing out that both LV and MV grids are exposed to the same risks since HFD can be transmitted from MV to LV grids and vice versa through power transformers [62], as mentioned in Section 2.3.

To mitigate HFDs, several potential solutions have been proposed in the scientific literature. Some of these solutions are summarized below:Filters, such as EMC (electromagnetic compatibility) filters, can mitigate the HFDs by creating a low-impedance path for high-frequency signals, thus preventing their propagation into LV and MV grids [68].Resonance control strategies can mitigate the HFD propagation [3].Impedance control strategies can mitigate HFDs by managing the impedance of the grid and connected devices to counteract the propagation [62].

Commonly used techniques in the literature deploy filtering schemes, which can have different characteristics:Passive filters are widely used for harmonic filtering and offer several advantages, including energy saving and reductions in power demand [69].Active filters are effective in compensating for current and voltage harmonics, reactive power, and voltage imbalance in three-phase systems [70].Distributed multiple low-voltage filters can be deployed in random locations of a distribution system to mitigate harmonic distortions [71].Passive zero-sequence harmonic filters can trap two harmonics with one filter and are effective in mitigating zero-sequence harmonics [72].

However, all these solutions have several unsolved issues related to the behavior of HFD, especially in MV grids. The lack of recommended practices to assess the effects of HFD on the power grids makes the diagnosis and the mitigation of the related problems challenging tasks. Moreover, as mentioned in Section 2.3, HFDs can propagate through transformers and the transfer ratio and amplification of different components can vary, making the propagation of HFDs less predictable [43]. Finally, HFDs can cause interference with elements for power delivery and end-user equipment, which complicates discovering the point of injection of these disturbances.

## 3. Standard Review

A considerable number of international standards on the topic of power grid disturbances have been published. However, there is not a specific international standard covering the disturbances in the frequency range of 9–150 kHz in an MV grid. So far, standards have only partially addressed voltage and current components resulting from disturbances and signaling within the frequency range of 9–150 kHz.

Considering the growing interest in HFD within LV and MV grids, numerous international technical committees (TCs) are actively revising existing standards and developing new standards to address HFD.

In this context, this section provides a comprehensive overview of the in-force international standards as well as TCs dealing with HFD. The relevant standards are categorized into four groups, as outlined in Table 1.

### 3.1. IEC and IEEE International Standards

In Section 2.2, the IEC 61000-4-7 [5] and IEC 61000-4-30 [20] standards were referenced as standards that provide methodologies for measuring voltage and current components up to 150 kHz. In addition to measurement methods, the IEC 61000-4-7 [5] and IEEE 1159 [73] also present a classification of power grid disturbances as conducted low-frequency and high-frequency phenomena. It is worth noting that the terms high frequency and low frequency in these standards are not defined in terms of a specific frequency threshold but instead indicate the relative difference in the frequency content of the phenomena listed in these categories. The main disturbances in the frequency range 9–150 kHz considered by the standards are the induced continuous wave (CW) voltage or currents, unidirectional transients, and oscillatory transients. However, the standards do not set specific limits to these disturbances for voltage and current in MV grids.

### 3.2. The IEC 61869-1 Edition 2 Standard

As outlined in the Section 1, the measurement of HFD in MV grids requires the use of ITs. Consequently, the frequency behavior of ITs significantly influences the accuracy associated with the measurement of these phenomena. In this respect, the IEC 61869-1 Edition 2 standard [27], which has been recently released, introduces new frequency accuracy class extensions for ITs up to 500 kHz. More specifically, it introduces five different extensions (WB0 to WB4) that differ in the covered frequency range. The WB0 class, which prescribes limits up to the 13th harmonic, is indicated as mandatory for low-power instrument transformers (LPIT) and SAMUs (standalone merging units), whereas the other extensions (WB1 to WB4) are optional. Regarding inductive IT, no wideband class extension is mandatory, not even the WB0.

However, the standard [27], but also the entire IEC 61869 standard family, does not provide guidelines on how to test ITs to verify their accuracy class at frequencies different from the power frequency (i.e., 50/60 Hz). Specifically, it does not specify the type of test signal to be used, i.e., whether the tests should be performed at reduced or nominal voltage. In this context, everything remains at the discretion of metrological institutes and testing laboratories intending to verify the accuracy of ITs according to the limited classes provided by the standard.

### 3.3. IEC Technical Committees

In recent decades, many technical committees (TCs) have focused their research on emissions within the frequency range up to 30 kHz. However, due to the escalating occurrence of emissions in recent years, primarily driven by the proliferation of high-frequency emission sources, TCs are now expanding their investigations to include emissions up to 150 kHz. The necessity of this extension is acknowledged by TCs, as depicted in Table 2. However, a considerable number of these TCs are standardizing LV grids and associated equipment for integration. The standardization of HFD, encompassing emission thresholds, measurement methodologies, sensors, transducers, and their metrological characterization for MV applications, remains an ongoing issue, with only a minority of TCs currently engaged in addressing it.

The document in [74], produced by IEC TC 8 as part of the IEC 63222 standard family, focuses on frequency-domain modeling for power quality analysis in electric power networks and provides a detailed list of the most common HFD-emitting installations (Table 3).

The TR [75] aims to provide guidelines for using power quality data to extract knowledge about power generation, transformation, distribution, and consumption aspects. Moreover, some specific applications are discussed, i.e., the analysis of additional losses of a transformer due to voltage deviations and harmonics and monitoring of the operation status of equipment (for instance, transformer overheating due to harmonics).

## 4. Previous European Research Projects Review

The traceable characterization of ITs, with reference to AC and DC grids’ applications, at frequencies different from the fundamental, currently represents an open issue for National Metrology Institutes (NMIs) and other metrological communities. Consequently, several European research projects have been funded in recent years to address this issue. This section aims to provide a concise review of previous research projects conducted over the years. For brevity, only key findings from these projects are presented here, and detailed information can be accessed through the referenced papers and webpages.

### 4.1. Smart Grid II

The EMRP ENG52 Smart Grid II [76,77] was funded with the main aim of implementing a comprehensive metrology framework for phasor measurement units (PMUs). Specifically, the project focused on meeting the accurate calibration requirements for PMUs and their associated transducers, crucial for monitoring the stability of distribution networks. Additionally, the project emphasized the need for innovative synchronized grid measurement techniques to analyze the propagation of PQ disturbances in networks, identify significant sources of disturbance, and measure network impedance.

In the framework of the European project Smart Grid II, an optimized, non-invasive CT was developed [78]. A non-invasive split-core CT with nanocrystalline core material was developed. Three different split-core CTs with primary currents of 100 A, 500 A, and 1000 A were designed and built. An extensive characterization was performed to quantify the CTs’ performance both in amplitude and frequency up to 5 kHz. The investigation of error frequency dependence revealed that employing nanocrystalline materials for CT core construction allows their utilization within a frequency range of up to 5 kHz, with ratio error variations on the order of a few hundred µA/A.

Within the Smart Grid II project, calibration facilities were developed for the calibration of inductive MV VTs under distorted realistic voltages up to 30 kV and frequency content lower than 7 kHz [78]. Specifications and constraints of the facility for the frequency-dependent calibration of MV inductive VTs were identified, considering a range of voltages from 1 kV up to 30 kV with superimposed harmonics up to the 50th. Two possible generation and measurement setups were identified and developed. The key elements of both setups are a power amplifier (30 kV, 20 mA, DC to 30 kHz, with full generation capabilities up to 7 kHz), supplied by an arbitrary waveform generator, and a suitably built digital comparator with associated control and measurement software.

The first proposed setup extended the one used for the two-step procedure based on a high-voltage (HV) capacitance bridge, which is currently employed in NMI laboratories for the calibration of inductive VTs at power frequency. The second setup, more oriented to the calibration of non-conventional VTs with LV output, was instead developed to calibrate VTs by comparison with a reference sensor. The reference sensor, designed and built for this particular purpose, is a resistive capacitive divider with suitable shields introduced to improve its frequency response and reduce proximity effects.

The first setup allowed for calibrating VT up to 10 kHz with standard uncertainty within 200 µV/V and 300 µrad for ratio and phase errors, respectively. The second setup produced similar uncertainties in a narrower frequency range (DC to 8 kHz).

Ratio and phase errors up to the first resonance were measured on commercial VTs, with primary voltages from 1 kV up to 50 kV. Measurements were performed carrying out a frequency sweep with the fundamental at the rated MV voltage (up to 30 kV), with a superimposed harmonic of amplitude from 20% to 0.2% of the fundamental component.

A technique was developed for compensating ratio and phase errors of MV VTs up to several tens of kHz [78]. This involved metrological characterization of the VT over a wide frequency range, followed by cascading a device with a frequency response equal to the transducer inverse transfer function to compensate for systematic deviations. An impulse infinite-response digital filter was adopted for this purpose, with its complex transfer function factorized in second-order sections. An objective function was defined to identify the best set of coefficients to fit the given transducer, using a hybrid scheme combining stochastic and deterministic approaches to minimize fitting errors. The technique was applied to compensate for the behavior of a 50 kV VT, resulting in significant improvement in its frequency response up to 5 kHz. The MATLAB implementation of the filter showed substantial enhancement in the frequency response, which was further implemented by using a PXI platform. Testing on a 55/√3 kV VT demonstrated compliance with standard limits up to 4 kHz, with a significant reduction in the resonance peak after compensation. Lastly, the compensating filter for a 1 kV VT was successfully implemented [79].

### 4.2. SupraEMI

The project 18NRM05 SupraEMI [80,81] dealt with the development of a metrological framework for the measurement of disturbances in the LV grids from 2 kHz to 150 kHz [82]. Within this project, a new measurement method for HFDs was proposed [24] and tested in the laboratory and onsite in different LV grids [83].

The new method was developed with the objective of updating the existing international standard IEC 61000-4-30 [20], which defines PQ measurement methods for power grids.

A major constraint on the development of the method was the need to demonstrate compatibility with a historic radio standard defined in CISPR 16-1-1 [23], which is used with an analogue radio receiver to sweep across the frequency band to look for emissions. This standard includes the 2–150 kHz band and, as such, has established itself in several EMC and PQ standards as the only available measurement method. For example, the basis of EMC is the definition of compatibility levels, which is the maximum level of interference on the electricity grid for which any connected appliance should ever expect to be exposed to. The compatibility levels must be measured and monitored in grids, but in the absence of a suitable measurement method, the committee in charge used to refer to [23], thus creating the constraint that a new practical method for grid measurements must respect.

The method had to resolve the amplitude of emissions in the 2–150 kHz range, which implies a decision on the frequency resolution for each component in the spectrum. The higher the resolution, the more data the instrument will produce, leading to issues of storage, data presentation, and interpretation. However, if the resolution is too low, it will not be possible to determine the emission problem frequencies, preventing the tracing of the source of the emission and its mitigation. As a compromise, a frequency resolution of 200 Hz was chosen. Computationally, calculating the 2–150 kHz magnitude spectrum is extremely demanding and the algorithm used needs to be as efficient as possible. Therefore, the processing requirements of the instrument are realistic within the low-cost constraints, which over the 2–150 kHz spectrum leads to 740 frequency results every measurement period.

Defining the measurement period (and time resolution) has similar constraints on storage and interpretation and is also fundamentally constrained by the selection of the frequency resolution. A basic measurement update time of 200 ms (every ~10 cycles of a 50 Hz grid) was selected, and aggregation over periods of 3 s and 10 min was suggested for compatibility level surveys to reduce storage and interpretation issues. The aggregation is obtained, for each frequency interval, by the square root of the square sum of the corresponding phasors, evaluated from every 200 ms intervals involved in the aggregation period.

The required accuracy of the measurement (including method and instrumentation) is 10% at each measured frequency for all magnitude values greater than 5% of the instrument range.

Emissions usually cause problems for equipment connected to the grid, which are generally considered in terms of the peak value and the root mean square (RMS) value of the emission. Therefore, the new method was conceived to be able to measure both peak and RMS of the individual frequency values across the spectrum. CISPR 16-1-1 already addresses these measurements but introduces a third type, called the quasi-peak value, which is based on an analogue circuit implementing a peak hold and decay function. IEC SC77A WG8 chose this quasi-peak value to be the basis of compatibility levels. Therefore, the new method had to produce this quasi-peak output too, using digital filters [82].

### 4.3. MyRailS

The main purpose of the EMPIR 16ENG04 MyRailS [83,84] was the development of both laboratory and on-board train calibration and measurement setups and robust data processing algorithms to enable high-accuracy energy and PQ measurements under highly dynamic electrical conditions. The envisaged frequencies ranged from a few hertz up to 5 kHz for AC systems and up to 3 kHz for DC systems. The uncertainty targets for laboratory calibrations were 0.5% and 0.1% for AC and DC systems, respectively, and 0.4% for on-board calibration of DC systems.

In this context, within the MyRailS project, a new system for the generation of high-voltage waveforms up to 25 kV–50 Hz and 15 kV–16.7 Hz, with harmonics up to 5 kHz and phase-fired current waveforms going up to 500 A, was developed [85]. In particular, the developed AC calibration facility can generate:A sinusoidal high voltage up to 25 kV, 50 Hz, or 15 kV 16.7 Hz, with up to 5 kHz harmonic content.A sinusoidal current (50 Hz or 16.7 Hz) up to a 500 A RMS value and up to 5 kHz harmonic content.

Moreover, in the framework of this project, a DC power measuring system was developed. It was able to generate DC power (phantom power) up to 8 MW with the supply voltage ranging from 1 kV to 4 kV and current from 10 A to 2000 A, and with a superimposed ripple with arbitrary frequency content up to 15 kHz for the voltage and 600 Hz for the current [86].

### 4.4. Future Grid II

The EMPIR project Future Grid II [86] provided missing solutions for the calibration and the timing of the digital substation instrumentation according to IEC 61850 [87]. The main objectives of the Future Grid II project were:Establish calibration methods to support testing of digital instrument transformers.Provide references for instruments with digital input or output.Develop tools for devices that exploit sampled values in digital substations.Create traceable references for the verification of time and synchronization methods.

In the framework of this project, a “universal” comparator was developed and characterized, able to compare the output of any type of transducer (voltage or current, conventional or non-conventional, analogue or digital) in the presence of PQ or PMU (intending here the typical voltage or current phenomena used to evaluate the accuracy of a PMU [88]) events with a frequency spectrum up to 9 kHz [88]. The developed “universal” comparator also included synchronization with GPS or a different sync pulse. Regarding the uncertainties, 100 µV/V (µA/A) for the voltage (current) magnitude and 150 μrad for the phase were the target uncertainties up to 9 kHz [89]. The universal comparator was included into the generation and measurement setup developed for the characterization of VT and low-power VT (LPVT) with digital output in the presence of PQ phenomena up to 9 kHz.

### 4.5. IT4PQ

Triggered by the needs expressed by the IEC TC 38, the primary goal of the EMPIR project 19NRM05 IT4PQ [28,29] was to establish a metrological framework that ensures the traceability of PQ measurements conducted by various types of IT, ranging from calibrations at NMIs to characterizations performed in industrial laboratories. To reach this objective, four key technical goals were pursued:Developing reference measuring systems and methodologies to establish reference systems for ITs and assess their contribution to PQ indices’ uncertainty.Establishing methods and procedures for calibrating ITs, including adherence to grid PQ disturbance limits outlined in current standards.Defining an “IT PQ Accuracy Class” for different types of ITs and PQ phenomena, based on identifying an overarching PQ performance index (PI) for ITs and its corresponding accuracy thresholds.Assessing IT behavior under realistic conditions, such as the simultaneous presence of various influencing factors.

In the framework of this project, a number of PQ phenomena with spectral content up to 9 kHz were analyzed, and the typical range of variation of each phenomenon was identified [90]. Suitable generation and measurement setups were implemented in order to test different types of ITs in the presence of PQ events [90]. In particular, reference systems were developed to assess the errors introduced by CTs, VTs, and combined ITs in PQ disturbance measurements. The general scheme of the developed setups includes a generation section, a reference sensor, and a suitable acquisition system. Utilizing programmable waveform generators and step-up transformers or power amplifiers, the generation systems are capable of generating stationary waveforms as well as transient PQ disturbances, such as dips, swells, and oscillatory transients, with a frequency content limited to 9 kHz. The applied waveforms are measured by characterized reference sensors, which may include MV resistive-capacitive dividers, VTs, CTs with associated conversion stages, and low-power CTs (LPCTs). Investigative efforts explored the potential use of a current comparator as an alternative reference sensor, demonstrating its feasibility. Digital bridges are employed to record outputs from both the reference and characterized ITs, with quantities of interest being evaluated either in the time or frequency domain. Standard measurement uncertainties are maintained within a few tens of µV/V and µA/A (µrad), respectively, for voltage and current ratio (phase) errors at power frequency, with a slight increase to a few hundred µV/V and µA/A (µrad), respectively, up to 9 kHz. As a main result, it was highlighted that the simultaneous presence of PQ phenomena impacts the performance of ITs, suggesting that to understand their actual metrological characteristics, it is necessary to test ITs in the presence of waveforms containing multiple disturbances. For instance, it was observed that for inductive VTs, the ratio error at the harmonic frequencies increased by an order of magnitude if the test signal included subharmonics in addition to harmonics.

The experimental setups were also used to investigate the impact of single- and multiple-influence quantities on ITs’ performance in the measurement of PQ disturbances. Several tests were performed under various conditions, such as temperature variations, burdens, vibrations, proximity effects, and adjacent phases. The primary finding revealed that burdens and temperature had the most significant impact on VTs, especially near the resonance frequencies. For LPITs, proximity effects and adjacent phases were found to be crucial. However, the way individual influencing factors interact with and influence IT performance largely depends on the specific type of IT and its construction.

## 5. Main Findings and Future Works

From the extensive review activity summarized in this paper, it can be concluded that HFD represents a quite recent topic of interest across various technical and scientific aspects of the electrical and electronic engineering.

The increased attention is driven, in turn, by the increasing complexity and sophistication of the modern electrical systems, which are more susceptible to such disturbances. They are often generated by power electronic converters and have significant implications for the quality and reliability of electrical power systems.

An analysis of literature concerning on-site measurement campaigns has shown that it is difficult to define a range of typical values for HFD in terms of frequency and amplitude. This difficulty arises due to its strong dependency on the characteristics of the emitting sources. Different types of power electronic devices, such as inverters, converters, and motor drives, exhibit distinct HFD profiles. It can be concluded that, in general, the frequencies of HFD are closely linked to the switching frequency of the converter generating the disturbance. For instance, a converter switching at 20 kHz may generate disturbances predominantly at that frequency and its harmonics. The amplitude of these disturbances can vary greatly, influenced by factors such as the network impedance, load conditions, and the design of the power electronic device itself. However, in the observed cases, the amplitude generally does not exceed 5% within the frequency range of 9 to 150 kHz.

By focusing on the metrological and normative context, it is evident that there are two important open issues:In-force Standards and Spectral Content Classification: Current standards focused on the quality of electrical power supplied by public distribution networks do not establish explicit limits in terms of amplitudes and occurrence for phenomena with spectral contents in the 9–150 kHz range. These standards primarily classify such phenomena without providing detailed guidelines or limits. This lack of regulation presents a challenge for maintaining consistent power quality across different regions and systems, potentially leading to varying levels of susceptibility to HFDs.Measurement Methodologies and Indices: On the other hand, standards addressing PQ disturbance measurements offer methodologies and indices for detecting and reporting HFD. However, these methodologies may not be comprehensive enough to cover all the scenarios and types of disturbances. Regarding the standards related to ITs, the Edition 2 of IEC 61869-1 [27] provides extensions of the accuracy classes up to 500 kHz, but it completely lacks information about methods, procedures, and test waveforms for the metrological characterization of ITs. This gap in standards means that, although ITs are a key element for accurate PQ assessments, there could be cases in which they may not be fully reliable for measuring high-frequency phenomena.

The analysis of the results of the concluded European metrology research projects has highlighted several issues in measuring high-frequency phenomena. These issues pertain to the adopted algorithms and indices. The performance of ITs, such as inductive MV VTs, in measuring high-frequency spectral components can introduce significant errors. For example, inductive MV VTs exhibit resonances starting from a few kHz with errors up to tens of percent, significantly impacting the accuracy of HFD measurements.

Furthermore, the necessity of testing ITs in the presence of realistic waveforms at MV combined with disturbances has been clearly demonstrated in several papers. Accurate testing procedures must replicate actual operating conditions, including the presence of the component at the grid rated voltage and frequency along with superimposed disturbances. This realistic approach ensures that the performance of ITs can be evaluated under conditions they will encounter in real-world applications.

Considering these main findings, the necessity of identifying suitable procedures and setups to verify the accuracy of ITs within the frequency range of 9–150 kHz is evident. Such procedures must include realistic test waveforms that reflect actual operational conditions, ensuring that ITs can accurately measure HFDs. It is reasonable, in general, to limit the amplitude of these disturbances to 5%, as higher amplitudes may not be representative of typical operational scenarios and could unduly stress the ITs.

Moreover, the development of accuracy indices for ITs based on various indices used for quantifying HFD is essential. These indices should provide a clear and consistent measure of IT performance in the presence of HFDs, facilitating better comparison and assessment across different devices and systems. By establishing this metrological infrastructure, characterized ITs can be deployed in power grids for the traceable measurement of HFDs. This deployment will offer critical data necessary for normative bodies and technical committees to enhance existing standards or develop new standards, ensuring better management and mitigation of HFDs in electrical power systems.

The ADMIT project [31,32] aims to comprehensively address these issues. By focusing on the development of accurate measurement techniques, realistic testing procedures, and robust standards, the ADMIT project seeks to enhance the capability of power systems to effectively handle HFDs. This initiative will not only improve the reliability and quality of electrical power but also support the ongoing integration of advanced technologies and renewable energy sources into the grid, which are known to introduce high-frequency disturbances.

To address the unresolved challenges in HFD measurement and mitigation, several future research directions and new technologies can be considered:Advanced Sensing Technologies: Developing more sensitive and accurate sensors for detecting high-frequency disturbances is crucial. Innovations in sensor materials and designs could enhance the detection capabilities for HFDs, providing higher measurement accuracy.Artificial Intelligence (AI) and Machine Learning (ML): The application of AI and ML algorithms can revolutionize the analysis and interpretation of HFD data. These technologies can help in identifying patterns, predicting occurrences, and distinguishing between different types of disturbances. ML models can be trained to recognize specific HFD signatures, aiding in quicker and more accurate diagnosis of power quality issues.Enhanced Signal Processing Techniques: Research into advanced signal processing methods, such as wavelet transforms and adaptive filtering, can improve the extraction and analysis of high-frequency components from electrical signals. These techniques can help in isolating HFDs from other noise and providing a clearer understanding of their characteristics.Development of New Standards and Regulations: Collaboration between research institutions, industry stakeholders, and regulatory bodies is essential for developing new standards that address the measurement and mitigation of HFDs. These standards should include clear guidelines on acceptable limits, measurement methodologies, and reporting practices, ensuring a unified approach to managing HFDs.Integrated Circuit Design Improvements: Enhancing the design of power electronic devices to minimize the generation of HFDs is another critical area of research. This could involve optimizing the switching frequencies, improving circuit layouts, and incorporating advanced filtering techniques to reduce the emission of high-frequency disturbances.Grid Modernization and Smart Grids: The transition to smart grids offers opportunities to integrate advanced monitoring and control systems capable of detecting and mitigating HFDs in real time. Smart grid technologies can facilitate dynamic adjustments to operating conditions, minimizing the impact of disturbances on the overall power quality.High-Frequency Filter Development: Designing and implementing high-frequency filters specifically tailored to the 9–150 kHz range can help in mitigating the effects of HFDs. These filters can be integrated into power electronic devices or deployed at critical points in the power network to suppress unwanted disturbances.Simulation, Modeling, and Digital Twins: Advanced simulation tools and models can provide valuable insights into the behavior of HFDs under various conditions. By simulating different scenarios, researchers can better understand the factors influencing HFD generation and propagation, guiding the development of more effective mitigation strategies.

By pursuing these research directions and embracing new technologies, the electrical and electronic research community can make significant strides in addressing the challenges posed by HFDs. This progress will enhance the reliability, efficiency, and quality of electrical power systems, supporting the continued growth and integration of advanced technologies and renewable energy sources.

## 6. Conclusions

This paper provided an overview of information on HFDs obtained from various sources, including scientific papers, international standards, and outputs from concluded European research projects. From the literature, it is evident that these phenomena are present in both LV and MV grids. However, identifying typical values of such disturbances is a challenging task since their frequencies and amplitudes strongly depend on the characteristics of various sources injecting them. From a metrological and normative point of view, various needs and open issues have emerged. Several algorithms and methods for measuring HFDs have been studied and proposed, each with their strengths and weaknesses. Regarding the ITs used upstream of the measuring instruments, the IEC 61869-1 standard [27] recognized the need to extend the concept of accuracy class up to 500 kHz but have left open the issue of how to verify such accuracy.

## Figures and Tables

**Table 1 sensors-24-04148-t001:** Relevant international standard families dealing with HFDs.

Reviewed International Standards	
Voltage characteristics of electricity supplied by public distribution networks	EN 50160 IEC 62749
Recommended practice for monitoring electric power quality	IEEE 1159
Electromagnetic compatibility (EMC)	IEC 61000
Instrument transformers	IEC 61869 IEC 60044

**Table 2 sensors-24-04148-t002:** IEC technical committees involved in the study of HFDs.

IEC Technical Committee	Scope
TC8-SC8A Grid integration of renewable energy generation	Standardization for grid integration of variable power generation from renewable sources, with emphasis on overall system aspects of grids.
TC 13 Electrical energy measurement and control	Standardization in the field of AC and/or DC electrical energy measurement and control, for smart metering equipment used in smart grids.
TC 17 High-voltage switchgear and control gear	Standardization of TS ^1^ and TR ^2^, covering high-voltage switchgear and control gear.
TC 38 Instrument transformers	Standardization in the field of AC and/or DC current and/or voltage instrument transformers.
TC 22-SC22F Power electronics for electrical transmission and distribution systems	Standardization of electronic power conversion and/or semiconductor switching equipment.
TC 51 Magnetic components, ferrite, and magnetic powder materials	Standardization of magnetic components, ferrite, and magnetic powder materials.
TC 77-SC77A EMC low-frequency phenomena	Standardization in the field of electromagnetic compatibility.
TC 85 Measuring equipment for electrical and electromagnetic quantities	Standardization of equipment, systems, and methods used in the fields of measurement, recurrent tests, monitoring, generation, and analysis of steady-state and dynamic electrical quantities.
TC 95 Measuring relays and protection equipment	Standardization of measuring relays, protection equipment, and protection functions.

^1^ TS Technical Specifications. ^2^ TR Technical Report.

**Table 3 sensors-24-04148-t003:** Types of HFD-emitting installations [74].

Installation Type Included	Power Quality Parameters to Be Concerned
Wind farm	Flicker, harmonics (inter-harmonics)
PV station	Harmonics, disturbances > 2 kHz
AC electrified railway	Unbalance, harmonics, voltage dip
DC electrified railway	Harmonics
AC electric arc furnace	Harmonics (inter-harmonics), flicker, unbalance
DC electric arc furnace	Harmonics, flicker
Induction heating furnace	Harmonics (inter-harmonics), flicker
Polysilicon ingot furnace, monocrystal oven, crucible oven	Harmonics
AC/DC rolling mill	Harmonics (inter-harmonics), flicker
Electric welding machine	Harmonics, flicker, unbalance
Electrolysis	Harmonics
Electric shovel, cargo lifter, and gantry crane	Harmonics
Adjustable speed drive (ASD)	Harmonics (inter-harmonics), flicker
Switched mode power supply	Harmonics
EV charger	Harmonics
Energy-efficient lighting	Harmonics

## Data Availability

Not applicable.

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
