# Peer review of "Power Grids and Instrument Transformers up to 150 kHz: A Review of Literature and Standards"

_sensors, 2024, doi:10.3390/s24134148_

Round 1

Reviewer 1 Report

Comments and Suggestions for Authors

The authors have reviewed the literature and standards relating to the issue of dynamic states of electrical networks and components of the measurement path in the frequency range 9 - 150 kHz. This frequency range is above the range of interest of standard relay protection devices and instrumentation, while these phenomena are particularly important for the assessment of power quality indices. The literature list contains a comprehensive collection of  articles and world documents relevant to the issues discussed.  The article also evaluates the latest methods related to the detection and analysis of electrical signals in the frequency range in question. This work can be a good introduction to research concerning the measurement of such signals and the identification of their sources, especially in low- and medium-voltage electrical networks.

Reviewer 2 Report

Comments and Suggestions for Authors

The research, standards, and literature on high-frequency distortions (HFD) in power grids are reviewed in detail in this paper. Of particular interest is the discussion of instrument transformers (ITs) and their accuracy up to 150 kHz. Scholars and professionals in the field who are interested in power quality issues associated with sophisticated grid technology and the integration of renewable energy sources will find great value in this thorough review.

In order to comprehend and lessen the effects of HFD in medium voltage (MV) and low voltage (LV) grids, it is essential to address the development of measurement standards and procedures, which are covered in full in this technically extensive review. The paper's ability to effectively connect theoretical concepts with real-world applications increases its relevance in these contexts.

The study is mostly a review of the literature; it does not include any actual data or particular case studies that show how the technologies and approaches under examination are applied or impactful in real-world situations. Such information could enhance the paper's contributions and give a more lucid depiction of the advantages or disadvantages of the standards and procedures in use today.

A more thorough discussion of potential future research directions or new technologies that could address unresolved challenges in HFD measurement and mitigation would be beneficial, even though the paper focuses on present standards and practices. This would open up a channel for continued study and development.

Include case studies or illustrations of the application of the standards and measurement methods that were described. This could make it easier to comprehend how the study's conclusions would apply in the actual world.

Give a more thorough explanation of any prospective standards or measuring technology developments that could improve the precision and dependability of HFD measurements. This can involve creating brand-new sensor technologies or machine learning and artificial intelligence-based prediction algorithms.

Extend the conversation about the drawbacks of the current approaches and offer solutions for future studies. To more clearly illustrate the advantages and disadvantages of each measurement approach, this could entail a comparative comparison of those techniques.

Comments on the Quality of English Language

Moderate editing of English language required.

Reviewer 3 Report

Comments and Suggestions for Authors

The paper provides an overview of scientific literature, standards, and the outputs of recently finished research projects dealing with the phenomenon of High-Frequency Distortion (HFD) in the electric grids, measurement techniques, propagation, and mitigation methods of HFDs for frequency range of up to 150 kHz. Also, Instrument Transformers (ITs) that are necessary for measuring HFD both at Low Voltage (LV) and Medium Voltage (MV) levels are analyzed. Moreover, a summary of the most common ranges of amplitude and frequency variations of actual HFD found in real grids is provided. Finally, the authors have also analyzed currently adopted measurement methods and the normative open challenges to be addressed. They have pointed out that current standards lack indications on test procedures, methods and specifications of the reference systems used for characterizing the ITs at frequencies different from the rated grid frequency.

Overall, it is an interesting topic that is presented most of the time clearly and concisely. However, there are certain issues with this paper that require a mandatory revision of the manuscript, so that the subject matter would be suitable for publication. Some of them are listed below:

#1 It appears that something is wrong with section numbering on pages 8 and 9. Where is section 3.1.1?

#2 Abbreviations CT, VT are not properly defined. Abbreviation PV is defined in rows 130 and 137.

#3 On page 9, row 432, the word “to” is probably missing in “up to 150 kHz”.

#4 It is common practice that references are numbered in the way they appear in the paper. Here this is not the case. For instance, references [16]-[20] first appear on page 5, after references up to [58] are previously introduced. There are other similar examples, and this is confusing for a reader. Thus, for the sake of clarity, I suggest complete renumbering of references.

#5 There are 229 references listed in the paper and some of them are not properly cited. As this is a review paper this must be corrected prior to publication. A reader should not use Google to search for and guess about references. Please, provide a working link or DOI for all recent journal and conference papers. For instance, what is reference [71],[78],[89],[121],[137],[138],[208]?

#6 References [106]-[111] differ in style from the rest of the list.

#7 Authors are omitted from reference [123]. Please check author names in references [54] and [95].

#8 Reference style is not consistent throughout the paper. As an example, please compare references in rows 66, 70, 71, 241, 317, 513, 563.

#9 On page 4, row 157, reference [41] is probably [50].

#10 On page 5, row 232, reference [16] should be [13].

#11 On page 12, row 513, I am not convinced that reference range [88-103] is correct.

#12 Finally, the major remark. There are just too many listed references never mentioned in the manuscript. What is their purpose? The authors should avoid self-citation if that is not necessary. Thus, all provided references should be properly cited in a revised manuscript.

Reviewer 4 Report

Comments and Suggestions for Authors

The paper is well written and the amount of literature reviewed for it is extensive. However, approx. 40% of the citations are actually self-citations, which in my opinion is quite a high ratio.

Being a review paper, not many comments can be given. I only have 2:

- the authors mention the negative effects of HFD on PQ and normal operation of the grid, but never state which are these negative effects. Please list them and state to what level of HFD the negative effects start to occur.

- please address the issue of the high number of self-citations.

Round 2

Reviewer 2 Report

Comments and Suggestions for Authors

All the comments have been incorporated. 

Author Response

Authors wish to thank the Reviewer.

Reviewer 3 Report

Comments and Suggestions for Authors

The total number of citations is reduced from 229 to 89. Listed relevant references are properly cited in a revised manuscript. The Authors have addressed the issue of unused references as well as the issue of self-citations, so now 10% of the citations are self-citations.

The revised paper is an improvement regarding the original manuscript and is suitable for publication.

Author Response

Authors wish to thank the Reviewer.

Reviewer 4 Report

Comments and Suggestions for Authors

The authors have not included my previous comments in their cover letter.

It is unclear how my comments were addressed.

Comments on the Quality of English Language

Quality of English is fine.

Author Response

Authors wish to thank the Reviewer.

Now the modifications with respect to the previous version were included in the cover letter.

Round 3

Reviewer 4 Report

Comments and Suggestions for Authors

My comments have now been addressed.

The paper has been improved and now it can be accepted for the publication.